# Investigating the welfare and conservation implications of alligator wrestling for American Alligators (*Alligator mississippiensis*)

**Casey Riordan, Jennifer Jacquet, Becca Franks**⊙*

Department of Environmental Studies, New York University, New York, New York, United States of America

* krf205@nyu.edu

**Data Availability Statement:** We are uploading all data as a supplementary material file.

**Funding:** CR and BF received an award from the Center from Environmental and Animal Protection (CEAP: https://wp.nyu.edu/ceap/) at New York

## Abstract

Wildlife tourism attractions (WTA) are popular in the United States, but they may be harmful to the individual animals involved and we question whether they provide benefits to environmental conservation. Most research on the welfare and environmental implications of WTAs focuses on charismatic mammals, with few studies investigating these issues for reptiles. Here we examine alligator wrestling, including its impact on animal welfare and environmental conservation. Using a sample of 94 relevant YouTube videos of alligator wrestling in Florida representing 16 different venues, we coded the environmental and behavioral characteristics evident in each video. We then performed a content analysis of wrestlers' narration in a subset of 51 videos to analyze the environmental awareness and educational components of alligator wrestling. Our results show systemic welfare harm: 11 venues housed adult alligators together with conspecifics, 96% of alligator wrestling performances facilitated direct contact in the form of physical restraint by one or more human wrestlers, and as many as 96% of the videos did not show a suitable water or waterside features for captive alligators. Furthermore, 12% of performances showed wrestlers flipping alligators onto their backs while 16% showed wrestlers tying alligators' jaws shut, both of which are known to be acute stressors. Finally, just under half of alligator wrestling commentary (49%) addressed environmental conservation topics, and much of this commentary included contradictory or misleading information that is not likely to benefit alligators in the wild. We argue that alligator wrestling serves no role in promoting positive relationships between humans, animals, and the environment, and instead furthers traditional notions of dominion that undermine welfare and conservation aims.

## Introduction

Wildlife tourism enables visitors to encounter or interact with non-domestic animals in captivity or in the wild [1] and is part of a lucrative recreation and amusement tourism sector [2]. Wildlife tourism attractions (WTAs) are visited by millions of domestic and international visitors each year [3]. Some scholars suggest WTAs are highly popular among the travel-

University. The funder had no role in study design, data collection and analysis, decision to publish, or preparation of the manuscript.

**Competing interests:** No authors have competing interests.

enthusiastic Millennial generation, who are expected to drive global tourism trends in the future [4–8].

Multiple categories of WTAs are described in the literature [1,9–16]. Attractions may take place within wild, captive, or semi-captive locations [1,9]. They may be considered consumptive or non-consumptive, depending on how the animals are used (for example, hunting versus bird-watching) [1,10–14]. Within non-consumptive WTAs, animals may be used for observational "encounters" in natural settings or for interactive "presentations," where they are displayed and often physically touched or handled to entertain tourists [1,15,16]. While WTAs and ecotourism are not synonymous, WTAs may be considered a form of ecotourism when they adopt and promote environmentally responsible practices [1]. Similar to ecotourist attractions, WTAs may offer economic benefits to local communities and support environmental conservation initiatives through volunteer work, donations, and education [11,17–22]. However, some WTAs explicitly advertise their benefit to environmental conservation even when these benefits do not exist; in one study by Moorhouse et al. (2015), 33% of captive dolphin encounters, 22% of wild dolphin encounters, 55% of elephant attractions, and 60% of shark cage diving attractions claimed in their publicity materials to support conservation despite an objective evaluation that demonstrated negative value to conservation [21,23].

A growing body of research also suggests that in practice, many WTAs have a negative impact on the individual animals involved [24]. For example, operators may place wild animals in artificial, monotonous environments that restrict their movement and ability to engage in natural behaviors [25]. Occasionally, animals may even be coerced into performing unnatural behaviors (i.e. elephant rides or bear dancing shows) or come into frequent, repetitive contact with unfamiliar human tourists [25–27]. For example, Jones (2015) describes an ecotourism attraction in Florida that illegally fed wild alligators to entertain airboat passengers, a practice that encourages alligators to approach humans and thereby increases their likelihood of being reported as a nuisance and killed by wildlife control officers [28,29].

Historically, studies addressing the welfare and conservation implications of WTAs have focused on charismatic mammals: e.g., elephants, big cats, and dolphins [30–35]. A few studies have examined WTAs involving marine turtles [10,36,37], but other reptiles are rarely addressed in the WTA literature. Meanwhile, hundreds of thousands of crocodiles, snakes, and other reptiles are housed in captivity and used in tourism attractions around the world [13,21].

The American Alligator (*Alligator mississippiensis*) is one example of a reptile used in WTAs in the United States. Alligators are large, carnivorous reptiles weighing anywhere from 90.7 to 226.8 kilograms and reaching up to 4 meters in length throughout their 50-year lifespans [38,39]. Native to the Southeastern Atlantic and Gulf Coast regions of North America, alligators prefer environments with both a land and a fresh or brackish water feature and are usually situated within swamps, creeks, rivers, and marshes that provide waterside foliage to hide from prey [40–42]. Adult alligators are solitary and territorial [43], although smaller individuals including females and juveniles often cohabit with other members of the species, also referred to as conspecifics [44]. Wild alligators have been known to attack humans, although this is rare [45]. Even so, alligators are often ranked along with crocodiles as a "least-favorite" or "fear-provoking" animal [46,47].

Culturally, alligators are most often associated with Florida. In 1987, the alligator was appointed Florida's official state reptile [48], and it is the mascot of the University of Florida. There are no data on the role of alligators in Florida's tourism industry, but the state is a leading national tourism destination, attracting over 112 million domestic tourists and 14 million international tourists in 2018 [49]. Alligators are housed in public zoos, farms, hunting

grounds, wildlife refuges, theme parks, and research centers throughout the state, and many venues allow tourists to safely hunt, view, or touch alligators.

A specific type of alligator tourism now known as "Alligator Wrestling" has historic origins. For thousands of years, Seminole Native Americans relied on alligators for their meat and hides [50], and some hunters developed manual capturing techniques that allowed them to mount, tie, and transport live alligators back to their camps for slaughter [51]. As Florida became a booming tourist destination in the early twentieth century, visitors paid to watch Seminoles capture alligators. Beginning in the 1900s, many Seminoles capitalized on this attention and shifted their use of alligators from hunting to wrestling, making an income off of flashy stunts during their performances [52]. For example, Seminole historian Frank (2012) claims alligator wrestlers would "[poke] and [prod] the gators. . .sit on the backs and stomachs of alligators, pry open their jaws and stick their hands and heads inside, use their hands and chins to keep the jaws shut, [and] drag and turn animals by the tip of their mouth" (p. 124) [51]. Today, alligator wrestling occurs in at least 13 venues around the state and recently received a surge of attention as a result of the *Animal Planet* series "Gator Boys," which ran from 2012 to 2015 and followed the exploits of a team of non-native alligator trappers and wrestlers [53].

Animal rights activists have become concerned with this WTA in recent years, leading to petitions from the Animal Rights Foundation of Florida, The Animal Rescue Site, and The Rainforest Site calling upon Florida's legislature to ban alligator wrestling [54–56]. Reptiles are capable of experiencing emotions such as anxiety, fear, distress, frustration, and suffering [57,58]. Wrestled alligators may thus be exposed to severe welfare threats from having their torsos and jaws restrained, being approached and handled by humans, and being flipped on their backs, all of which have been shown to be acute stressors [59–64]. Furthermore, like other captive reptiles, wrestled alligators may be housed in artificial enclosures with groups of unfamiliar conspecifics, a systemic stressor for many reptiles in captivity [59,65,66].

While the case may be made that for native wrestlers, alligator wrestling is a component of their history, some non-native attractions instead emphasize alligator wrestling as a way to promote environmental conservation education [67–72]. For example, some attractions claim on their websites and social channels that they are conservation nonprofits, that they collaborate on conservation research or donate money to crocodilian conservation efforts, or that they use their alligators to train conservation officers in proper animal handling [67–70]. Others claim to educate tourists about alligator conservation during their alligator wrestling performances without specifying what tourists are meant to learn and how their education benefits alligators and the environment [71,72]. As a result, the conservation benefits of alligator wrestling remain unclear. WTAs such as the St. Augustine Alligator Farm also claim to support environmental conservation initiatives without offering alligator wrestling performances, further questioning whether wrestling is a necessary means to achieve conservation goals.

Some alligator wrestling attractions also claim to support alligator conservation by providing "sanctuary" to unwanted or nuisance alligators reported by members of the public. In the state of Florida, it is illegal to relocate a nuisance alligator trapped in the wild because relocated alligators often return to their original site or threaten alligators in their new location [73]. Instead, alligators deemed a nuisance due to their size and location near humans and domestic animals are often slaughtered, with 7,669 nuisance alligators harvested out of 14,072 permits issued for alligator removals in 2019 [74]. Wrestling attractions that use only nuisance alligators thus argue that they benefit alligator conservation by preventing nuisance alligators from being killed by trappers in the wild [75]. However, the International Union for the Conservation of Nature (IUCN) lists the conservation status of alligators as "least concern" [76], a

designation that suggests keeping alligators in captivity is currently unnecessary to directly sustain overall population levels [21]. And while the IUCN lists captive alligator research as a recommended conservation action for the species [76], it is unclear whether studying alligators used in wrestling matches, where they are potentially exposed to unnatural encounters and environmental conditions, would produce insights that can accurately inform captive and wild population management. Taking in nuisance alligators to prevent them from being harvested in the wild could instead be argued as a benefit from an animal welfare perspective, but the conditions inflicted upon alligators used for wrestling indicate that this is perhaps not the case.

Here we examined alligator wrestling in Florida with the overarching aim of investigating and evaluating the animal welfare and environmental conservation claims made at these attractions. We created a database of wrestling performances filmed by audience members who subsequently posted their videos to the social media website YouTube. In recent years, social media platforms have become important sources for collecting data on animal welfare and behavior, WTAs, and human-animal conflicts [77–81]. We developed and applied a coding scheme to document the presence and absence of various environmental conditions and human behaviors. Finally, we conducted a content analysis of wrestlers' monologues in a subset of videos included in our database to assess whether these attractions provide environmental conservation education or another value to environmental conservation initiatives.

## Materials and methods

### Ethics statement

We collected all of our data through YouTube using only publicly available data published under the standard YouTube license or Creative Commons CC BY license. The use of our dataset was done in compliance with the YouTube Terms of Service.

### Database creation

From June 3 to December 1, 2019, we systematically searched YouTube for videos using the terms "alligator wrestling" and "gator wrestling" (as the term 'gator' is often used colloquially for alligators). Because some wrestling attractions refer to their alligator wrestling by a different name, we also used each attraction as an individual search term along with the word alligator (for example, "Croc Encounters"+"alligator"). Finally, the Florida Fish & Wildlife Conservation Commission (FWC) provided a list of all facilities and individuals with a Class II license to possess and display alligators in the state of Florida, so we included each facility as a YouTube search term to ensure we had accounted for every entity in legal possession of an alligator throughout the state.

Our YouTube search revealed hundreds of alligator wrestling videos, but we aimed to assess only captive alligator wrestling taking place in Florida. Therefore, we eliminated videos that showed alligator wrestling in the wild, those that claimed to be alligator wrestling but were actually something else (for example human-to-human wrestling or a pet playing with a stuffed toy), and videos where the location was not in Florida. To deduce the location of each video, we used the video title and description or referred to signage shown in the actual footage. This procedure yielded 94 videos for analysis from 13 different Florida-based attractions that are currently in operation, as well as two attractions that are no longer operating (noted as such in Table 1) and seven videos categorized under 'Unnamed Florida Location' because the facility names were not discernible anywhere in the videos. The earliest video was posted on April 13, 2007, and the most recent was posted on March 28, 2019. This number was not exhaustive, but it allowed us to assess a comprehensive amount of footage from numerous native and non-native facilities. The list of videos used can be found in S1 Dataset.

**Table 1. Characteristics of YouTube videos in the dataset.**

| Location | Total Videos | Percentage | Median Year | Year Range | Median Video Length (in seconds) | Median Views |
|---|---|---|---|---|---|---|
| Everglades Holiday Park | 25 | 26.6 | 2013 | 11 | 275 | 7024 |
| Gatorland | 22 | 23.4 | 2016 | 8 | 435 | 439 |
| Miccosukee Indian Village | 7 | 7.4 | 2016 | 9 | 539 | 1136 |
| Unnamed FL Locations | 7 | 7.4 | 2015 | 8 | 207 | 4101 |
| Gator Boys Road Show | 5 | 5.3 | 2014 | 6 | 207 | 23425 |
| Jungle Queen | 5 | 5.3 | 2016 | 9 | 286 | 296 |
| Native Village[a] | 4 | 4.3 | 2013 | 1 | 171 | 5530 |
| Billie Swamp Safari | 3 | 3.2 | 2015 | 9 | 527 | 3333 |
| Everglades Alligator Farm | 3 | 3.2 | 2014 | 5 | 341 | 25749 |
| Kachunga Alligator Show | 3 | 3.2 | 2013 | 8 | 182 | 302 |
| Miami Gator Park | 3 | 3.2 | 2008 | 5 | 150 | 1638 |
| Captain Jack's | 2 | 2.1 | 2016 | 1 | 85 | 4533 |
| Gator Adventure Productions | 2 | 2.1 | 2012 | 6 | 106 | 368 |
| Croc Encounters | 1 | 1.1 | 2013 | 1 | 188 | 710 |
| Gator Golf Adventure Park | 1 | 1.1 | 2018 | 1 | 371 | 69 |
| Native Village Roadshow[a] | 1 | 1.1 | 2016 | 1 | 346 | 199 |
| **Total** | **94** | **100** | **2014** | **11** | **296** | **1956** |

[a]*Attractions are no longer in operation.*

## Environmental and behavioral coding

To accurately assess the environmental and behavioral characteristics of alligator wrestling, we watched each YouTube video twice. In the first round, we familiarized ourselves with the behaviors performed in alligator wrestling and noted descriptive information about the videos including the upload date, number of views, length of video in seconds, and wrestling location, summarized in Table 1. We also coded for characteristics of each performance arena (the location within each attraction where the alligator wrestling takes place). Specifically, we coded for habitat characteristics that are known to be important to alligators in the wild [41,42,82], including whether there was water evident in the arena; whether the water was clear or murky; whether the water was deep enough for alligators to submerge their bodies; and whether there was waterside foliage present. We also noted whether alligators were isolated or housed with other alligators in the performance arenas, and the size of the arenas on an ordinal scale (small, medium, or large). We defined "large arenas" as those with at least 28sqm of space for the alligator within the enclosure. "Medium arenas" provided a more restricted area of approximately 11 to 28sqm of room to move. "Small arenas" were those with approximately 11 sqm of room or less. To classify arena dimensions, we first calculated the mean length of an adult alligator at 3 meters, based on the average length of female alligators (2.6 meters) and male alligators (3.4 meters) [83]. From there, we then estimated the arena sizes in relation to the body length of alligators in each enclosure.

In the second round of views, we coded for alligator wrestling behaviors, including how many humans participated in a performance (determined by the number of humans who physically came into contact with an alligator during the video) and the number of wrestled alligators (determined by the number of alligators who were physically touched by human(s)). We noted whether wrestlers physically restrained alligators' legs and torsos, the duration of restraint, and whether the video captured the entire restraint duration from start to finish. We also looked at the wrestling stunts mentioned by historians as traditional features of alligator

wrestling, including poking or slapping an alligator on the jaws; pulling an alligator's jaws open; sticking one's head and hands inside an alligator's mouth; using one's chin to keep the alligator's jaws shut; and dragging an alligator by the tip of the mouth [51]. Finally, we coded for other behaviors based on their repeated occurrence in the videos, specifically when a wrestler used their chin to hold an alligator's jaws open; when they dragged an alligator by the tail; when they taped an alligator's jaws shut; and when they poked an alligator in the eyes.

We assessed descriptive statistics for the data, noting the percentage of videos that included each habitat characteristic and each wrestling stunt described above. We also calculated the percentage breakdown of human and alligator wrestlers and the median amount of time in seconds that alligators' legs and torsos were restrained in each video, starting from the moment when a wrestler mounted the back of an alligator and stopping when the wrestler was no longer touching the alligator with any part of his or her body. We conducted all statistics on SPSS Statistical Software Version 26.

## Content analysis

To examine the claims of environmental conservation benefits of alligator wrestling, it was necessary to get a better understanding of the wrestlers' monologues during their performances. As such, we conducted a content analysis of a sub-sample of the data set. To be considered for this portion of the analysis, videos had to exceed one minute in length; they had to include discernible dialogue (for example, videos that contained only music or were too muffled to discern from start to finish were excluded); they had to show footage from a standard alligator wrestling performance for the public (for example, videos showing a private wrestling demonstration without an audience were excluded); and they had to take place at a non-native-operated attraction. Although we chose to include native facilities in our quantitative analysis, we excluded them from the content analysis under the assumption that native facilities can use alligator wrestling to promote their local culture and history, which has been cited on numerous occasions as an acceptable justification for WTAs [23,28,84]. Though we did not consider native attractions for this portion of the study, it is important to note that having minimum animal welfare and conservation standards should be important ethical considerations for all wrestling attractions moving forward, especially amid the increasing global environmental concerns. We ultimately transcribed 51 videos (54% of total data set) for this portion of the study.

We carried out our content analysis manually, without using a qualitative software package. Although this approach may limit the depth of analysis that is possible for large datasets [85], we felt that the limited size of our dataset and the nature of our research questions were such that a rigorous analysis could be carried out manually [86]. To complete the coding process, we read through each video transcript and noted whether the video addressed environmental conservation issues or another topic. Afterward we assessed the monologue more closely, grouping wrestlers' narration into the relevant theme or sub-theme according to the topic discussed. In addition to environmental conservation topics, we found two other categories of wrestler monologue that were prevalent throughout the data: narratives about "entertainment" and "miscellaneous education." As such, our analysis included a combination of deductive and inductive coding [87]. All themes and sub-themes are summarized in Table 2 and discussed further in the results section of the study.

## Quantitative results

### Arena characteristics

Regarding arena size, we found that all performance arenas allowed alligators enough space to turn around, but room for movement varied by attraction. Fifty-five percent of performance

**Table 2. Summary of content analysis themes and sub-themes.**

| Theme | Description | Percentage of Videos |
|---|---|---|
| Theme 1: Environmental Awareness | Wrestler comments on issues about environmental conservation and/or protection | 49.0 |
| Sub-theme 1.1: Wildlife Cohabitation | Wrestler comments on living and interacting with wild alligators | 80.0 |
| Sub-theme 1.2: Alligator protection | Wrestler comments that their venue supports alligator rescue and rehabilitation | 60.0 |
| Theme 2: Education | Wrestler attempts to educate tourists about something other than environmental conservation/protection | 94.1 |
| Sub-theme 2.1: Alligator anatomy | Wrestler discusses aspects of an alligator's physical anatomy | 100.0 |
| Sub-theme 2.2: Culture/History | Wrestler addresses the Seminole origins of alligator wrestling | 43.8 |
| Theme 3: Entertainment | Wrestler makes comments purely for entertainment with no educational benefits | 96.0 |
| Sub-theme 3.1: Alligator jokes | Wrestler makes a joke about alligators | 57.1 |
| Sub-theme 3.2: Other jokes | Wrestler makes a joke about something other than alligators | 81.6 |
| Sub-theme 3.3: Wrestling instructions | Wrestler explains how to wrestle alligator | 81.6 |

arenas could be classified as "large" (28 sqm of room) while 25% of arenas could be classified as "medium" (11 to 28 sqm of room). Only 19% of arenas were classified as "small," meaning alligators had a restricted space to move of 11 sqm or less.

In the majority of videos (89%), water could be seen in the performance arena. This is perhaps unsurprising, as Florida policy requires facilities housing captive alligators to provide both a land and a water feature with enough space for animals to move and turn within both features [88]. However, among arenas that included a water feature, 14% did not appear to be deep enough for alligators to fully submerge themselves, while 42% appeared to contain murky or non-clear water and only 4% contained waterside foliage, features that alligators prefer in the wild [42]. Finally, despite the fact that alligators tend to be solitary animals, 67% of alligators were wrestled in an arena that also held conspecifics, while only 17% were in an arena where conspecifics were not evident. The remaining 16% of arenas could not be classified, either because the videos contained clips of several different wrestling performances or because the water feature was too murky to discern other alligators.

## Wrestling behaviors

We also assessed behaviors that occurred during alligator wrestling. The range of wrestled alligators used in performances was anywhere from one to three, but the majority of videos showed only one wrestled alligator (79%). The number of human wrestlers ranged from one to five, but similar to the number of alligators, most videos showed only one human wrestler (68%) and only 2% of videos showed four or more wrestlers.

Physical restraint was extremely common throughout the videos, with 96% of wrestlers physically restraining the legs and torsos of wrestled alligators. Most wrestlers restrained only one alligator (84%), while 5% of wrestlers restrained two or more alligators, and the remainder of cases (11%) were unclear. In the 90 videos where restraint was present, the median restraint time was 120 seconds (two minutes) but selecting only the 25 videos (27%) that showed an entire restraint period from start to finish, the median restraint time increased to 300 seconds (five minutes).

Many of the wrestling stunts mentioned as traditional in the historical literature [50–52] were not prevalent in the videos. For example, we never observed a wrestler mounting an alligator's stomach, and just over a third of videos showed wrestlers poking, prodding, or slapping alligators on the jaws (38%). Only 10% of wrestlers dragged alligators around the arena by their jaws, 30% stuck their hands in the alligators' open mouths, and 5% put their heads in the

alligators' open mouths. Finally, only 12% of videos showed wrestlers flipping alligators onto their backs. Other uncommon characteristics that were not mentioned in previous literature included videos showing wrestlers taping wrestled alligators' jaws shut (20%) and wrestlers poking alligators in their eye sockets (12%). Two percent of videos showed wrestlers dragging alligators with nooses.

When pulling alligators around the performance arena, the majority of wrestlers dragged the animals by their tails (61%). The most frequent stunts performed in the videos were the "Florida smile," the "face-off," and the "bulldog" techniques, illustrated in Figs 1–3. During the Florida smile, evident in 61% of videos, a wrestler forcibly pulls the alligator's jaws open, preventing the animal from closing its mouth. Bulldogging was present in slightly less than half of the data (49%) and occurs when a wrestler uses only his or her chin and chest to hold an alligator's jaws shut. Both the Florida smile and bulldogging were mentioned by Frank (2012) as wrestling stunts [51]. On the other hand, the face-off, which occurred in 52% of videos, has not been described in the historic literature. This occurred when a wrestler used only his or her chin to keep the alligator's jaws open.

## Qualitative results

Through a content analysis of 51 videos from the original dataset, we set out to analyze comments made by wrestlers during performances at non-native attractions. In less than half of the videos, wrestlers addressed topics related directly to environmental conservation and awareness (Theme 1, 49% of videos). More commonly, the videos contained commentary that intended to educate tourists about miscellaneous topics not directly related to conservation (Theme 2, 94% of videos) or focused on pure entertainment (Theme 3, 96% of videos). Because wrestlers often addressed more than one theme and/or sub-theme in their commentary, the percentages throughout this section do not add up to 100%.

### Theme 1: Environmental awareness

Commentary about environmental awareness topics fell into two sub-themes: comments about cohabiting with wild alligators (80% of videos in Theme 1) and comments about alligator protection (60% of videos in Theme 1). Regarding alligator cohabitation, some commentators discussed how to run away from an alligator. For example, commentators from at least one attraction regularly debunked the urban legend that running in a zig-zag motion is the best way to escape an alligator: "Yeah, you run straight, and you run fast, [but] don't run zig-zag. It's a myth, it's a total lie. [Alligators] only run like seven miles an hour. If you can't outrun that, trip your friend and walk. That'll work too" [70]. Other wrestlers discussed how to handle being attacked or bitten by an alligator, with several advising viewers to "go for the nose" or cover the alligator's eyes.

Other commentators shared less defensive, more practical information about encountering and cohabiting with alligators in the wild. Rather than focusing on an alligator attack (an unlikely event), these wrestlers instead provided conservation-driven messaging advising people to respectfully avoid alligators. For example, some commentators encouraged audience members to keep their distance, to avoid natural bodies of water where alligators may be present, and to understand that alligators naturally fear humans. One wrestler also shared information about Florida's nuisance alligator protocol in several videos, attempting to explain that alligators deemed a nuisance must be killed or placed into captivity, and can no longer live in the wild. This wrestler (and others from the same attraction) highlighted that alligator trappers are unpaid employees who receive the slaughtered nuisance alligator's meat and hide as payment, thus making nuisance reports a death sentence for wild alligators [25]. Finally, several

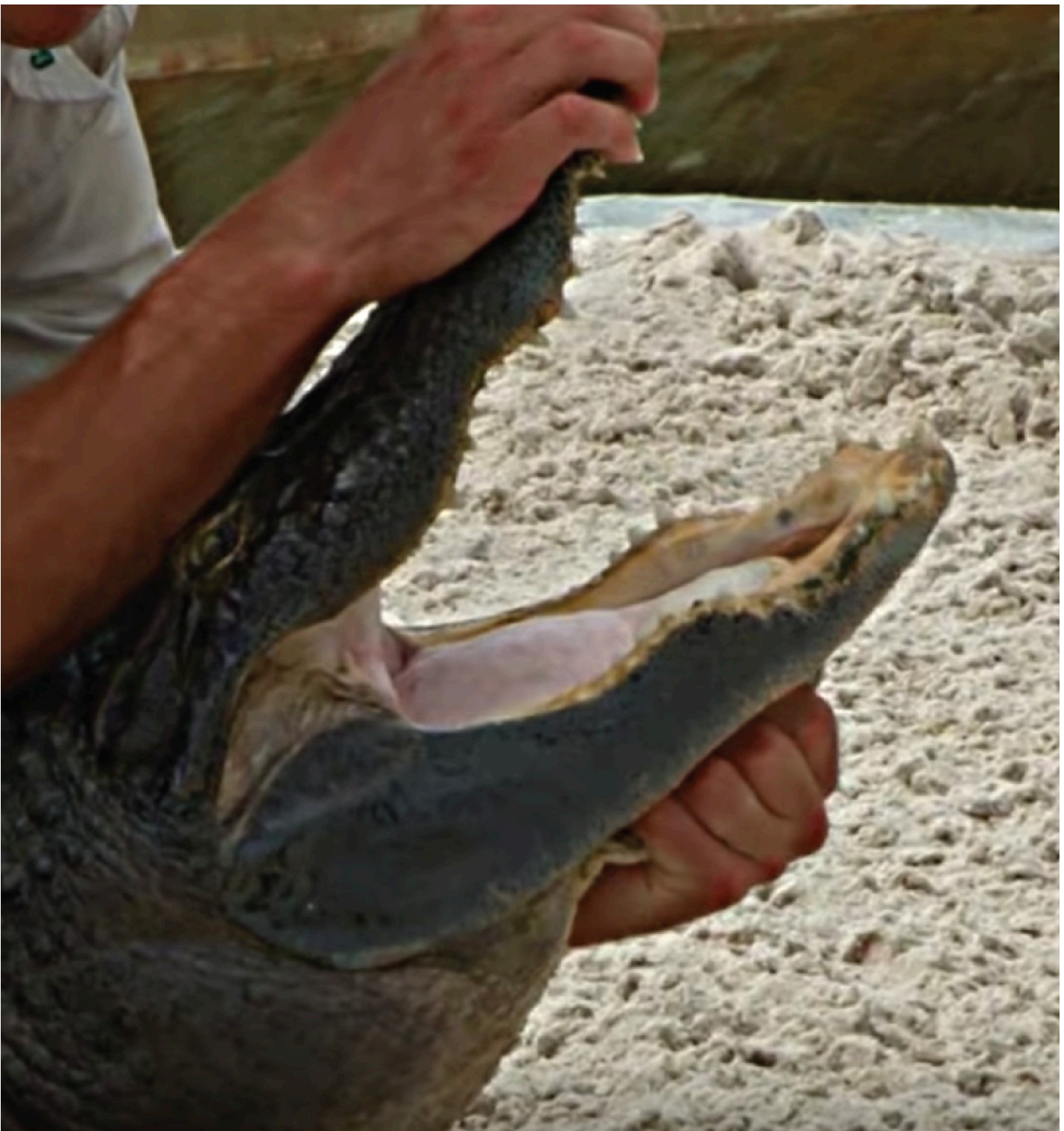

**Fig 1. The "Florida Smile" stunt.** Image captured from a video in the dataset.

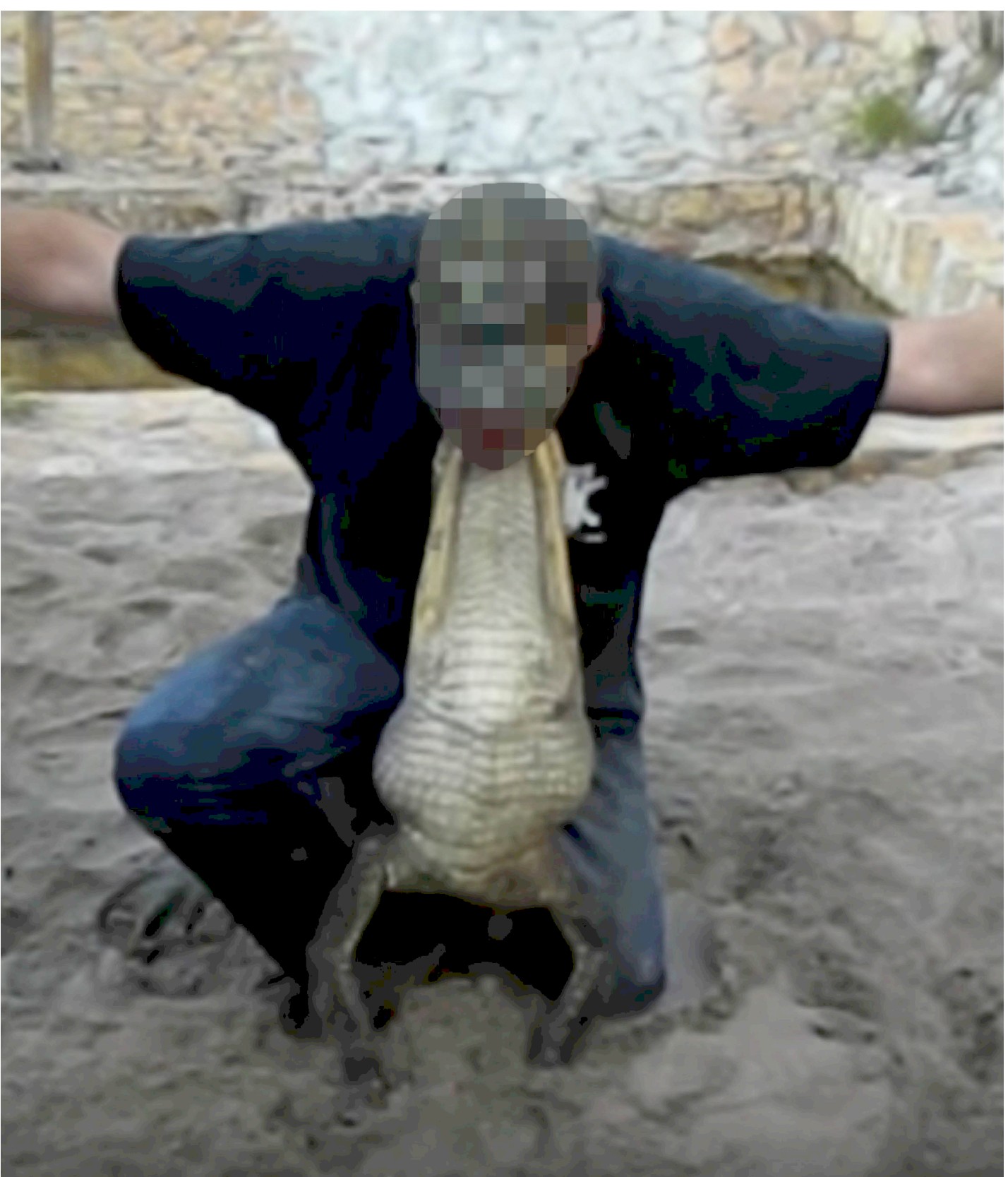

**Fig 2. The "Bulldogging" stunt.** Image captured from a video in the dataset.

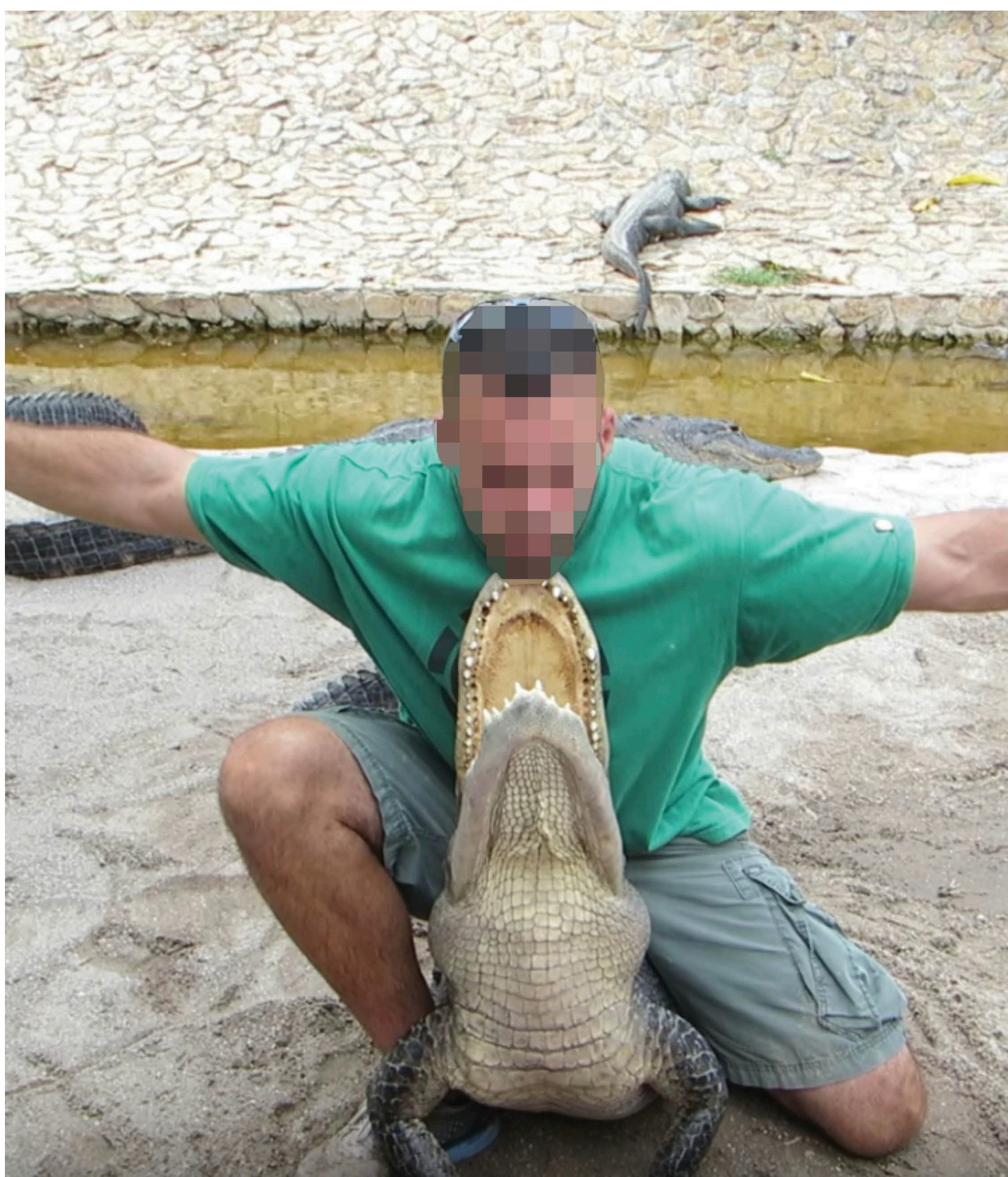

**Fig 3. The "Face-off" stunt.** Image captured from a video in the dataset.

wrestlers told visitors to avoid feeding wild alligators, including one who explained: "Don't be a nuisance person. Don't feed [an alligator in the wild]. Once you do, he'll learn to associate people with food and then if you come back without the food, you guys might become the food, which is what we want to avoid" [33].

The second sub-theme under environmental awareness was alligator protection. For example, some wrestlers stated that their facility rescues and rehabilitates wild nuisance alligators or informed the audience that their donations support the upkeep of captive alligators in their venue. However, given the treatment of alligators who are repeatedly wrestled at these facilities, it is unclear whether rescuing nuisance alligators from the wild and placing them into captive wrestling venues is more preferable from a welfare standpoint than killing nuisance alligators, which is the standard protocol set forth in Florida's nuisance alligator management program [73]. Furthermore, from a conservation perspective, not one wrestler explicitly addressed how their rescue and rehabilitation efforts support overall alligator conservation goals. This is important because alligators were previously listed as an endangered species from 1967 until 1987, when populations recovered largely due to successful conservation programs [89]. Though wild populations are currently thriving enough to make captive management arguably unnecessary to directly sustain overall population levels [21], maintaining captive alligator populations may also help to mitigate against future declines if their captive maintenance incorporates opportunities for the natural behaviors and experiences that would be necessary for population survival. However, without specifying the actions that alligator wrestling attractions are taking to sustainably and responsibly manage their alligators in accordance with statewide conservation goals, the conservation value of these attractions remains unclear. Finally, none of the commentators noted whether their alligator wrestling attraction makes a monetary donation toward external conservation or protection initiatives, which may be another way to benefit ongoing conservation efforts.

## Theme 2: Miscellaneous education

In 94% of the dataset, wrestlers attempted to educate tourists about miscellaneous topics that were not directly related to alligator protection or conservation. This type of commentary fell into two sub-themes: comments about alligator anatomy (100% of videos in Theme 2) and comments about the culture and history of alligator wrestling (44% of videos in Theme 2). These types of comments may arguably be useful in other ways, for example by presenting opportunities for wrestlers to discuss more conservation-focused topics; however, we coded these types of comments under "miscellaneous education" because they are not directly focused on conservation education.

Regarding alligator anatomy, the most common anatomical features discussed in the commentary were the alligator's metabolism, tail, nose and breathing ability, visual blind spots, anatomical differences from crocodiles, brain, length and weight, scutes (bony plates on the alligator's back), ears, skin color, eyes (including eye sockets and eyelids), jaws, throat, bite pressure, and teeth. The teeth were the most common feature mentioned, as wrestlers usually performed the "Florida smile" stunt to show the inside of the alligator's mouth and discuss the teeth. For example: "The alligator has about 80 teeth, 40 on top, 40 on the bottom. . .Their teeth are like a shark's, they're always growing, so what you see there is a cap. If they break a teeth off [sic], they'll grow one right back there in its place" [31].

Wrestlers who discussed the culture and history of alligator wrestling either mentioned the Native American origins of the activity or explained that Seminoles used bulldogging as a method for a single hunter to capture live alligators and bring them home. Notably, there were some contradictions in the historical information provided by wrestlers. Commentators at one

attraction repeatedly claimed that alligator wrestling was created by cattle hands who wrestled alligators during cattle drives to prevent them from eating cows seeking water on the route [i.e. 90]. We were unable to find a factual basis to this claim. Furthermore, whether or not it is true, many scholars of ecotourism highlight the importance of respecting and promoting local cultures through WTAs, rather than eliminating them from the story as was the case at this attraction [23,28,84].

### Theme 3: Entertainment commentary

In 96% of the dataset, wrestlers used their performances as an opportunity to entertain audiences through jokes and humorous commentary. This type of commentary fell into three sub-themes: jokes about alligators (57.1% of videos in Theme 3), other jokes (81.6% of videos in Theme 3), and wrestling instructions (81.6% of videos in Theme 3).

When jokes were made at the expense of alligators, some wrestlers went as far as to manipulate a wrestled alligator's body to make their audience laugh. For example, several wrestlers mentioned an alligator's ability to sink its eyeballs into its eye sockets while poking a wrestled alligator in the eyes and making noises such as "boop!" and "blurp!" [i.e. 76,83,91]. Other wrestlers would ask an alligator questions while holding and nodding its jaws, as if the animal was nodding yes [i.e. 16]. Wrestlers from one attraction could be seen flipping wrestled alligators onto their backs and pretending to punch them in the stomach as a regular part of their scripted performance [i.e. 70]. Finally, while mounted on an alligator, one wrestler grabbed its back leg and twirled it around in the dirt, telling the audience: "you can even play tic tac toe with him in the sand" [33].

In addition to manipulating an alligator's body, wrestlers also made light of an alligator's defense mechanisms or aggressive reputation. While making these jokes, the commentators usually anthropomorphized alligators to make them appear less frightening to the audience. For example, in one video, a wrestled alligator could be seen wiggling and thrashing underneath a human wrestler. The wrestler looked at the alligator and said: "He likes to make it look dangerous sometimes. It's a bit of a show. . .follow the script!" [23]. In another video, the wrestler restrained an alligator attempting to escape from underneath him and said: "He just wanted to come up [to the audience] and say hi" [52]. Some alligator jokes took on a sexual tone, such as an announcer who interrupted a wrestler in the middle of the "bulldog" stunt to warn him about getting a hickey (a blemish caused by kissing or sucking the skin) [6]. In multiple videos filmed at one attraction, a wrestler commented that he and his wrestled alligators "cuddle together in bed at night" [12].

Alligators were not the only ones subjected to wrestler humor, as other jokes included gender-based commentary and comments about the wrestlers themselves. Regarding gender, some wrestlers stated that their wrestled alligator must be a male, because if it were a female, they "wouldn't be able to keep [the alligator's] mouth shut" [i.e. 58]. Others noted that relocated alligators who return to their original location are like an ex-girlfriend [i.e. 12], and similarly one announcer commented that a wrestler likes to name all of his alligators after ex-girlfriends [79]. Even when the jokes were made about wrestlers rather than alligators, these jokes often played upon conventional notions of gender. For example, announcers at one venue could often be heard teasing male wrestlers for being weaker than the wrestled alligators [i.e. 5,70]. Many wrestlers also remarked on their own lack of intelligence or accomplishments, for example by claiming they are the "only. . .animal in the entire world. . .[with] a smaller brain to body size ratio [than alligators]" or that becoming a wrestler requires "a big stick, a low IQ, and good health insurance" [25].

Finally, wrestlers often instructed the audience about the act of wrestling an alligator during their performances. We classified this type of commentary as entertainment, rather than education, because there is nothing inherently beneficial from an environmental, historical, cultural, or scientific perspective about teaching the public how to wrestle an alligator. For example, wrestlers often talked about wearing an alligator out prior to a wrestling performance, catching an alligator, jumping on its back, securing its jaws, and getting off at the end of a performance. In these instances, the wrestlers would usually narrate what they were doing without emphasizing, for example, the historical significance behind their actions or the anatomical features of an alligator that required them to perform certain wrestling behaviors.

## Discussion

In this study, we found that alligator wrestling attractions may be causing systemic welfare harms to the alligators involved with few, if any, environmental conservation payoffs. Historically, alligators and other reptiles have received far less attention in the scientific literature than mammals, and knowledge of their cognitive and behavioral responses to stressors is still fairly limited [57,92,93]. Given the large number of alligators kept in captivity for alligator wrestling, the role of alligator wrestling in the evolution of Florida's tourism industry, and its supposed links to environmental conservation and education, it is surprising that this WTA has been largely neglected in the animal welfare, conservation, and tourism literature.

Only two prior studies have explicitly addressed the contemporary alligator wrestling industry: the first examined tourist perceptions of crocodile and alligator wrestling to show how tourist enjoyment is often prioritized over the animals' experiences [91], and the other described a cranial injury sustained by a human wrestler without discussing the performance itself or the animal(s) involved [94]. To address this knowledge gap, this study sought to provide the first empirical overview of alligator wrestling in Florida. In addition to revealing the typical behaviors and venue characteristics of wrestling matches, the study aimed to consider the animal welfare implications of the individual animals involved in wrestling as well as the environmental conservation benefits of this type of WTA.

### Alligator wrestling and animal welfare

Existing evidence suggests that alligators may experience stress or discomfort when they are flipped onto their backs [59], when their jaws are tied shut [64], and when they are manually restrained by humans [63]. An additional study found that physically restraining crocodilians is more stressful for them than using a stun gun, and that it can take up to eight hours for their blood stress levels to reduce after being physically restrained between two and 15 minutes [60]. Our study revealed that 96% of wrestlers restrained alligators in their performances, and that the median restraint time for videos that showed the entire duration of restraint was five minutes. This indicates that many wrestled alligators are being restrained long enough to heighten their blood stress levels. Furthermore, although flipping alligators on their backs and taping their jaws shut was not a common behavior seen in the data, these actions were still performed in 12% and 20% of videos, respectively. This finding suggests that at least some wrestlers may be causing repeated acute welfare damage to the alligators used for their performances.

Several of the most common alligator wrestling stunts seen in our data set have yet to be studied as stressors in the literature. Specifically, it is unclear whether the "Florida smile" (present in 61% of videos), "Face-off" (52% of videos) and "Bulldogging" (49% of videos) stunts are harmful to alligators. However, given that wrestlers physically restrain alligators and forcibly manipulate their bodies to perform these stunts, it can be assumed that they are at least somewhat stressful for the animals involved.

In addition to the potentially stressful behaviors inflicted upon wrestled alligators, it is likely that some of the environmental features present in the performance arenas may pose welfare threats to wrestled alligators. For example, Florida law requires licensed facilities to provide captive alligators with water deep enough to fully submerge themselves [88], but 14% of water features present in the data set did not appear to meet this criterion. It is worth noting that some wrestlers bring wrestled alligators into a temporary arena for performances, after which they are released back into a larger maintenance enclosure with water features that meet the requirements set out by law. Based on the limited nature of YouTube data, this study was only able to consider performance arenas, and while certain performance arenas appeared to double as maintenance enclosures, it was unclear the extent to which this was the case in the data set. Further research is needed to evaluate the environmental features in maintenance enclosures compared to performance arenas. Additionally, some videos showed travelling reptile exhibits, which are exempt from the standard caging requirements and permitted to temporarily house reptiles in smaller enclosures that meet health and safety needs but do not necessarily meet the standard legal caging requirements [90].

Furthermore, 42% of arenas contained features with clear water, whereas research suggests alligators prefer murky water in the wild [42]. Finally, alligators are solitary animals [43], but more than half of the wrestling attractions appeared to house wrestled alligators with conspecifics in the performance arenas. It is true that smaller alligators tend to cohabit in groups, which is perhaps why the alligators in many wrestling attractions appeared to be living peacefully with others in a single enclosure; however, smaller alligators also prefer habitats with waterside vegetation to conceal themselves [42], and only 4% of attractions provided such foliage in their arenas.

Finally, research shows that a major stressor for both captive and wild alligators is being approached and contacted by humans [60,61,63]. In this study, nearly every video showed wrestled alligators being physically restrained by human wrestlers, and every wrestled alligator in our data set was also surrounded by a human audience. Enduring human contact is not unique to wrestled alligators, as it is a common stressor imposed upon nearly every animal kept in captivity at wildlife tourist attractions, even those with strict animal welfare standards [25]. However, given the added stressors experienced by alligators who are repeatedly used for wrestling performances, it may be of particular importance for all wrestled alligators to have (at a minimum) access to murky water or waterside vegetation to conceal themselves similarly to how they would in the wild.

## Alligator wrestling and environmental conservation

In a study used to assess the conservation benefit of WTAs around the world, Moorhouse et al. (2015) emphasized that venues housing captive animals who are not endangered in the wild usually provide few conservation benefits, unless they offer an "indirect conservation benefit" such as educating tourists (p. 5) [21]. Our analysis of wrestlers' commentary found that 94% of commentary focused on educational topics not directly related to conservation efforts, such as alligator anatomy and the cultural or historical origin of wrestling. Even so, promoting the cultural dimension of alligator wrestling is understandable, as scholars generally agree on the importance of using ecotourism attractions to highlight marginalized local communities and traditions [95,96]. Furthermore, discussing the origins of wrestling and the anatomical features of alligators may arguably be a way for wrestlers to engage their audience before introducing conservation or welfare discussion topics. However, most wrestlers in our data set did not go on to discuss more direct conservation and protection topics, as just 49% of the videos we analyzed contained commentary about environmental conservation and protection issues. And

while the argument may be made that teaching people about alligators can promote empathy and long-term willingness to protect these animals in the wild, some scholars have found that anatomical information is less memorable for tourists than providing practical tips to support conservation initiatives [10,11].

Some video commentary did feature practical advice about human-alligator cohabitation, which may support conservation efforts. This was especially the case for wrestlers who told their audience not to feed wild alligators, and to leave them alone when encountering one in the wild. However, a number of wrestlers instead played into the stereotypical human fear of alligators by telling tourists how to run away from a wild alligator or how to fight back when attacked. Although it may be helpful to know how to outrun an alligator in the wild, prioritizing this information perpetuates the misconception that alligators are always threatening in the wild. In reality, alligators typically fear humans and only approach them when they have been fed by people in the past; experts argue instead that being actively chased by an alligator is rare, and that it is best to safely keep one's distance from these animals in the wild [97]. Data from the FWC reveals there were more than 17,000 nuisance alligator complaints and 14,000 permits issued to remove nuisance alligators in 2019 [74]; although several wrestlers at one attraction mentioned the state nuisance alligator protocol and its consequences for individual alligators, it would be beneficial for more wrestlers to discuss the circumstances that necessitate reporting a wild alligator as a threat.

Some wrestlers also focused on the importance of rescuing nuisance alligators from the wild or emphasized that tourist donations are used to support the captive alligators at their attraction. While this is perhaps beneficial from an animal welfare perspective, these comments failed to differentiate welfare from conservation. A similar phenomenon was highlighted by Moorhouse et al. (2015), who discovered that captive dolphin, elephant, and shark attractions sometimes claimed to benefit welfare and conservation without differentiating these terms, and usually benefitted neither welfare nor conservation [21]. The authors argued that such an attempt is "highly likely" to mislead and attract tourists to an inhumane attraction [21,23].

Furthermore, even if alligator wrestling attractions do provide genuine conservation value, wrestlers failed to adequately address how the captive maintenance of wrestled alligators supports overall conservation goals. Ballantyne et al. (2009) note the importance of communicating the reasoning behind management practices, especially ones that may be misleading such as asking tourists to stay away from alligators in the wild while simultaneously wrestling them in captivity or manipulating and mocking alligators' bodies for the sake of a joke (something that even the most responsible wrestlers in the data set tended to do). Such misleading information can confuse tourists, ultimately detracting from the attraction's overall educational value [10]. It also leaves room for public misinterpretation about the alligators and their needs, something Louw (2006) referred to as the danger of speaking on behalf of wild animals [98].

Of course, this is not to say that the WTAs incorporated in this study are purposely attempting to fool tourists. Indeed, one study of tourists attending WTAs in central Florida, including one alligator wrestling attraction, found that many tourists believed these attractions benefitted environmental conservation and were justified in existing for the purpose of educating tourists about the environment [99]. However, from an ethical standpoint, claiming to promote the conservation of alligators without making it evident to visitors what actions are being taken prevents tourists from fully understanding how alligator wrestling attractions (and how their monetary contributions toward these attractions) benefit conservation efforts. Furthermore, even if alligator wrestlers make a more concerted effort to educate tourists with more accurate and explicit conservation messaging, we question whether the act of wrestling animals in itself is the most effective way to promote positive and humane relationships between people and the environment. Instead, by glorifying the subjugation of alligators, these venues propagate

traditional notions of humankind's dominion over the environment and our right to interfere with the lives of wild animals who live in our surroundings, regardless of the threats we pose to individual animals and the environment in the process.

## Future research

We chose YouTube as our data collection source for several reasons. First, videos on YouTube can be accessed remotely at any time of day, and it is free to post and view YouTube content. Therefore it was more efficient in terms of cost and time to view alligator wrestling online versus physically traveling to each venue and watching dozens of alligator wrestling performances in person. Furthermore, collecting data through social media allowed us to avoid biasing alligator wrestlers who might alter their performances based on our presence [79]. Finally, studying wrestling performances that already happened and were available online enabled us to avoid the ethical implications of paying to support an activity that is likely to be inflicting harm upon captive alligators.

Of course, with any social media study there are limitations to the data collection. First, we relied on videos that were available and accessible online through our chosen keywords, limiting our sample. Second, many of the videos only showed a portion of an alligator wrestling performance or edited one or multiple performances into small clips. Though 25% of videos showed the entire duration of restraint in each wrestling performance, only 6 videos in the data set (6%) showed the full performance from start to finish including the wrestler's introductory and concluding comments to the audience. We accounted for this in our coding process, but it is possible that some of the wrestling stunts and commentary may have been excluded from these videos. Similarly, our perspective of alligator wrestling was restricted to what was shown on screen, meaning it was not always possible to assess the entire performance arena and its features in the videos. Finally, several of the videos used in the content analysis were difficult to discern at parts due to external noises, making the commentary slightly muffled. However, this was only the case for a minority of videos.

While this study has offered an empirical foundation of alligator wrestling and has assessed some of its implications for alligator welfare and conservation, future scholars may consider building upon this study to get a deeper understanding of alligator wrestling and its contribution to the growing field of reptile ethology. Scholars may consider interviewing audience members to assess how alligator wrestling narratives improve upon visitors' environmental knowledge, attitude toward alligators, or willingness to support alligators and the environment. Longitudinal methods can also be used to assess the salience of these narratives over time. Such studies may be used to improve the messaging used at alligator wrestling attractions as well as other WTAs that claim to focus on environmental conservation education. Moving forward, as tourism scholars continue to reveal the many detrimental welfare and environmental impacts of the wildlife tourism industry, it is important to consider other examples of WTAs that use lesser-studied species and taxa that are similarly exploited for tourism without environmental conservation tradeoffs.

## Supporting information

**S1 Dataset.**
(XLSX)

## Author Contributions

**Conceptualization:** Casey Riordan, Jennifer Jacquet, Becca Franks.

**Data curation:** Casey Riordan, Becca Franks.

**Formal analysis:** Casey Riordan, Becca Franks.

**Funding acquisition:** Casey Riordan, Becca Franks.

**Investigation:** Casey Riordan, Becca Franks.

**Methodology:** Casey Riordan, Becca Franks.

**Project administration:** Casey Riordan, Becca Franks.

**Resources:** Casey Riordan, Becca Franks.

**Software:** Casey Riordan, Becca Franks.

**Supervision:** Becca Franks.

**Validation:** Casey Riordan, Becca Franks.

**Visualization:** Casey Riordan, Becca Franks.

**Writing – original draft:** Casey Riordan, Becca Franks.

**Writing – review & editing:** Casey Riordan, Jennifer Jacquet, Becca Franks.

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
