## [Decision Letter · Decision Letter 0]

17 Sep 2020

PONE-D-20-17349

Investigating the welfare and conservation implications of alligator wrestling for American Alligators (Alligatormississippiensis)

PLOS ONE

Dear Dr. Franks,

Thank you for submitting your manuscript to PLOS ONE. After careful consideration, we feel that it has merit but does not fully meet PLOS ONE’s publication criteria as it currently stands. Therefore, we invite you to submit a revised version of the manuscript that addresses the points raised during the review process.

We look forward to receiving your revised manuscript.

Kind regards,

Bi-Song Yue, Ph.D

Academic Editor

PLOS ONE

Journal Requirements:

2. Please provide a statement indicating that the use of this Youtube dataset was done in compliance to the Youtube Terms and Conditions and our requirements for this type of study (https://journals.plos.org/plosone/s/submission-guidelines#loc-personal-data-from-third-party-sources).

Reviewers' comments:

Reviewer's Responses to Questions

**Comments to the Author**

1. Is the manuscript technically sound, and do the data support the conclusions?

Reviewer #1: Partly

Reviewer #2: Yes

Reviewer #3: Yes

2. Has the statistical analysis been performed appropriately and rigorously? 

Reviewer #1: Yes

Reviewer #2: Yes

Reviewer #3: Yes

3. Have the authors made all data underlying the findings in their manuscript fully available?

Reviewer #1: Yes

Reviewer #2: Yes

Reviewer #3: Yes

4. Is the manuscript presented in an intelligible fashion and written in standard English?

Reviewer #1: Yes

Reviewer #2: Yes

Reviewer #3: Yes

5. Review Comments to the Author

Reviewer #1: Investigating the welfare and conservation implications of alligator wrestling for American alligators (Alligator mississipiensis)

Authors analysed videos of alligator wrestling from YouTube and evaluated welfare and conservation matters. Results showed that the reptiles presented a low welfare and that no conservation message is being passed by the wrestlers. The paper is very interesting, brings novelty to the scientific literature, and proposes a modern method for this type of study. For this, the study should be considered for publication. However, some revision in needed. See the commentaries bellow:

Line 54: insert a reference after 'literature'

Line 72: insert a reference after 'behaviors'

Line 85: insert a reference after 'United States'

Line 126: insert a reference after 'education'

Line 140: insert a reference after 'facility'

Line 194: provide the criteria to classify each venue as large, medium or small. This criteria is already in lines 248-254, but it is better to remove them to line 194.

Lines 261-263: what about the 16% remainder cases?

Lines 439-440: but the mean wrestling time was 5 minutes and the majority of the wrestkers are not performing the stressing behaviors.

Lines 451-467: it is important to separate the maintenance enclosures from the show enclosures. To keep more animals together in the show enclosure is more attractive for the audience. An evaluation of the maintenance enclosures showed be conducted to see if the animals are being hold together. This, in fact, was suggested by the authors at the 'Future research' topic.

Lines 468-476: the total time of exposure to humans should be considered to evaluate if the animals are suffering from acute or chronic stress.

Line 496: insert the year of the study of 'Moorhouse et al.'

Lines 550-558: It is a good study method, but it can input errors in the analysis, as atated by the authors, due to the small duration of the videos. How many videos had the entire recording of the wrestiling show? Please, inform this here.

Line 578: to interview the audience could bring information about the conservation message learned during the shows.

Reviewer #2: Wildlife tourism attractions (WTA) with alligators are popular in Florida but the authors report that only two research studies have previously investigated these activities. The present study therefore wants to fill this gap. The analysis took into consideration 94 YouTube videos to investigate what happens during alligator wrestling and to collect data that can give indications about the welfare of the animals involved and the conservation implications of such activities. The authors found that in WTA classified as "alligator wrestling" animals are subjected to interactions that do not guarantee their welfare. Furthermore, the analysis of the contents of the wrestlers’ narrations revealed that they are mainly for recreational purposes and only superficially deal with conservation issues. The manuscript is technically sound, and the data support the conclusions. The position of the authors is therefore justified when they say that even if wrestling activities were accompanied by valid educational and conservationist messages, they would still have doubts that wrestling in itself can be an effective way to promote a positive relationship between animals and humans. It is therefore ethically sound that the authors have chosen a data collection method that does not feed these types of activities.

Below are some suggestions that might clarify and deepen some passages of the manuscript.

Lines 86, 250, 252, 254. Pounds, feet, and yards should be replaced with the units of the international system of Units (SI).

Line 143 Not all authors agree that keeping alligators in captivity is useless to protect wild populations. Among the conservation actions reported in the IUCN Red List is included the following paragraph. “Because of the extensive commercial ranching and farming industry in the USA, the American alligator is a prime candidate for research on captive husbandry. Incubation and rearing techniques need to be improved to increase the efficiency of the alligator ranching industry. Extensive research on these topics is currently underway, particularly in Louisiana (Reigh and Williams 2016) and Florida.” (Elsey, R., Woodward, A. & Balaguera-Reina, S.A. 2019. Alligator mississippiensis. The IUCN Red List of Threatened Species 2019: e.T46583A3009637. https://dx.doi.org/10.2305/IUCN.UK.2019-2.RLTS.T46583A3009637.en.

Line 229 - 232 Although native facilities have cultural and historical justifications, from an ethical point of view, this is not sufficient to derogate from minimum animal welfare standards. Moreover, given the increasingly severe environmental problems throughout the planet, the use of animals should always be accompanied by a commitment to conservation and the promotion of respect for nature.

Line 250. It is not clear what the enclosure size is. It would seem that "large" is an enclosure of at least 28 sqm (a circle with a radius of 4.57 m), "medium" from 11 to 28 sqm; "small" less than 11 sqm. It is also not clear how the authors arrived at this estimate.

Lines 325 and 332. Instructions on how to avoid contact with wild alligators and not to feed them are messages aimed at improving interactions between humans and animals. Therefore, not only can they be considered conservation messages, but since they are explicit indications on which behaviors to adopt or not to adopt, they respond to the indications provided by the study of Ballantyne et al. (2019) (see comments below for lines 488 -491).

Line 342 The fact that a species is classified as a least concern does not guarantee that the species will not decline in the future even in the short term. There are examples in the history of widespread species that have been extinct in a short time (e.g., Ectopistes migratorius). The same species Alligator mississippiensis was classified as endangered in the 1960s. Today the species is not at risk because of the success of the management and conservation plans implemented. It could be an example of sustainable management despite withdrawals.

Line 347 and line 516 Talking about the anatomy of the species could be a way to introduce animal welfare issues, so it could still be a useful message even if not strictly related to species conservation.

Lines 471 - 473 Although it is undeniable that zoo animals may be subject to some stress due to noise, public proximity, or other factors. When comparing animals housed in zoos and those used in wrestling with alligators, it would be right to also highlight the differences between the two types of management.

Line 488- 491 Quoting Ballantyne et al. (2009), authors affirm that “Among wrestlers who discussed environmental topics, none discussed the role of alligators in their natural habitats, which has been cited as an important component of conservation education as it helps visitors understand why protecting these animals is needed”. But it seems that the core message of the article cited is different: “Tourists were particularly interested in practical information about what they could do to help protect the wildlife, rather than general information about conservation issues.” (pag. 663) and “The findings of this study suggest that the key to balancing the needs of tourists with the needs of wildlife, is to clearly communicate the reasons behind particular management practices in terms that relate directly to protecting the animals from human impacts.” (pag.663)

Lines 540 - 543 Furthermore, from an ethical point of view, claiming to promote the conservation of the species without clarifying what actions are taken by the various venues is also a limitation for visitors. In fact, visitors do not have the necessary information to understand whether the economic contribution they make by paying for the ticket really benefits conservation and how.

Reviewer #3: Abstract

Line 30: Remove ‘of’ between ‘94’ and ‘YouTube’

Line 31: Why ‘at least 15 venues’? Can you be more accurate? Later, in table 1 (line 197), 16 venues are listed.

Otherwise, a very good abstract.

Introduction:

This section describes categories of WTA and potential impacts on the wildlife before introducing the understudied alligator and the history of wrestling. Final paragraph explains what the study has done. Here an explanation of the overarching aim could be more clearly made. As reference is made later to numbered research questions, see below, are they missing from here?

Materials and Methods:

Line 186: “To answer research questions 1 and 2 ..” The reader has not yet been told what these are (see comment connected to the missing aim from introduction).

Line 205: Alligator in this sentence should be singular – remove the ‘s’.

Content Analysis:

Line 233: the reader is told here that the transcription happened ‘by hand’. What does this actually mean? Date of the transcriptions is included here but does not seem relevant as findings should not change if transcription happened at another time. Relevant is when filming of the videos took place, which is information you have provided.

Line 235: Similar to above, the reader is told that the analysis happened ‘by hand’. What does this mean? Presumably you mean a software package, such as NVivo, was not utilized. It may be useful to explain why you choose to identify themes manually.

Quantitative results:

Line 249: Check journal requirements, but it is not usual to start a sentence with a percentage written in numbers (eg ‘55%’ here). This will need checking throughout the manuscript.

Line 250: ‘gators’ rather than ‘alligators’ used here with no apparent reason for the difference. This also occurs elsewhere on the manuscript (eg line 267 ‘alligators’ but line 268 ‘gators’ – in the same sentence). Suggest using one term (alligators) consistently throughout the text.

Line 292: preventing the animal from ‘slamming’ its mouth shut, or from closing it at all? Is the issue here the ‘slam’ (if not, this word seems inappropriate)?

Qualitative results:

Line 308: the figure of 94% comes from anatomy and/or culture (ie as two different categories combined, such that each performance may only have one of the two but still be counted here) or where both anatomy and culture were mentioned in the same performance? Later (lines 347-348) 94% is cited just for anatomy and 41% for culture … confusing this a bit further. Clarity is needed.

Line 342-343: Unclear sentence. Needs rewording.

Line 389: Safe to assume your international audience know what a ‘hickey’ is?

Discussion:

The numbered research questions referred to in the methods section are not revisited (at least not with reference to numbers) in the discussion. Conclusion is strong and relevant to the findings presented, but needs basing in the aim/research questions which should have been clearly identified at the outset.

Overall:

The content is very interesting and the manuscript well written.

A definition of ‘conspecific’ might be warranted, rather than assuming all readers will be familiar with this term that is used often in the paper. Lines 261-263 describe this without explaining at that point that conspecifics is the topic?

‘Stunts’ and ‘tricks’ seem to be used interchangeably throughout the manuscript. Are they the same thing, or different? If the former, I suggest using just one term consistently throughout. If the later, then an explanation of the difference to avoid confusion would be beneficial.

Similarly, sometimes ‘alligator wrestling matches’ is used and sometimes more simply ‘alligator wrestling’. Are they the same thing, or different? If the former, use one term consistently throughout. If the later, then explain the difference to avoid possible confusion.

The major issue to clarify (see above) is the research questions.

6. PLOS authors have the option to publish the peer review history of their article (what does this mean?). If published, this will include your full peer review and any attached files.

Reviewer #1: No

Reviewer #2: No

Reviewer #3: **Yes: **Georgette Leah Burns

---

## [Author Response · Author response to Decision Letter 0]

8 Oct 2020

Please see the attached document, Response to Reviewers, for detailed information on how we modified our MS in response to the reviewer questions and suggestions.

---

## [Editor Report · Decision Letter 1]

27 Oct 2020

Investigating the welfare and conservation implications of alligator wrestling for American Alligators (Alligatormississippiensis)

PONE-D-20-17349R1

Dear Dr. Franks,

We’re pleased to inform you that your manuscript has been judged scientifically suitable for publication and will be formally accepted for publication once it meets all outstanding technical requirements.

Kind regards,

Bi-Song Yue, Ph.D

Academic Editor

PLOS ONE

---

## [Editor Report · Acceptance letter]

6 Nov 2020

PONE-D-20-17349R1 

Investigating the welfare and conservation implications of alligator wrestling for American Alligators (*Alligator mississippiensis*) 

Dear Dr. Franks:

I'm pleased to inform you that your manuscript has been deemed suitable for publication in PLOS ONE. Congratulations! Your manuscript is now with our production department. 

Kind regards, 

on behalf of

Dr. Bi-Song Yue 

Academic Editor

PLOS ONE